# Whole-Genome Amplification—Surveying Yield, Reproducibility, and Heterozygous Balance, Reported by STR-Targeting MIPs

**DOI:** 10.3390/ijms23116161

**Published:** 2022-05-31

**Authors:** Ofir Raz, Liming Tao, Tamir Biezuner, Tzipy Marx, Yaara Neumeier, Narek Tumanyan, Ehud Shapiro

**Affiliations:** Department of Computer Science and Applied Mathematics, Weizmann Institute of Science, Rehovot 761001, Israel; ofir.raz@weizmann.ac.il (O.R.); taoliming.too@gmail.com (L.T.); tcom111@gmail.com (T.B.); tzipy.marx@weizmann.ac.il (T.M.); yaara.neumeier@weizmann.ac.il (Y.N.); narek.tumanyan@weizmann.ac.il (N.T.)

**Keywords:** whole gemome amplification, single cell genomics, short tandem repeats

## Abstract

Whole-genome amplification is a crucial first step in nearly all single-cell genomic analyses, with the following steps focused on its products. Bias and variance caused by the whole-genome amplification process add numerous challenges to the world of single-cell genomics. Short tandem repeats are sensitive genomic markers used widely in population genetics, forensics, and retrospective lineage tracing. A previous evaluation of common whole-genome amplification targeting ~1000 non-autosomal short tandem repeat loci is extended here to ~12,000 loci across the entire genome via duplex molecular inversion probes. Other than its improved scale and reduced noise, this system detects an abundance of heterogeneous short tandem repeat loci, allowing the allelic balance to be reported. We show here that while the best overall yield is obtained using RepliG-SC, the maximum uniformity between alleles and reproducibility across cells are maximized by Ampli1, rendering it the best candidate for the comparative heterozygous analysis of single-cell genomes.

## 1. Introduction

Single-cell genomes reveal cellular heterogeneity and allow the identification of somatic genomic changes such as point mutations, indels, copy number variations (CNVs), and others. Such changes play a key role in elucidating fundamental concepts in biology and medicine, such as the origin and developmental process of cancer, developmental relationships between cell types, and cell turnover in regenerating tissues [1]. Following the development of single-cell genomics technologies, increasing numbers of modules are being integrated in single-cell analysis, such as transcriptomics, chromatin accessibility, epigenetics, and protein markers [2]. Somatic mutations combined with transcriptomics in single-cell resolution [3] can help link the function of mutations discovered through genome-wide association studies (GWAS) to the key cell types. Phylogenetic relations among single cells can add a developmental layer on top of other single-cell profiles and can help investigate the transitions of cell states in certain conditions such as cancer evolution and embryonic development. Cell lineage discovery in humans is based on de novo mutations that occur in individual cells during each cell division. Obtaining these de novo mutations in single cells remains challenging due to the imperfections of whole-genome amplifications, loss of coverage, uneven amplification, and artificial noise introduced during amplification. In a recent example, Breuss et al. reported on the sequencing of RepliG-SC whole-genome amplification (WGA) products at 300× for the detection of somatic mutations^6^, stressing the need for balanced allelic representation.

Available WGA protocols cover several approaches that differ by various parameters [4], namely multiple displacement amplification (MDA), which provides good coverage but poor uniformity, and polymerase chain reaction (PCR)-based approaches, which offer better uniformity and reproducibility but lack in coverage. The thermostable polymerases used in the PCR-based approaches also exhibit higher error rates than phi29 used in isothermal MDA protocols [5,6]. Given the abundance of commercially available WGA kits, it is not clear which one best suits the sensitive analysis of heterozygous loci across the genome, such as a lineage discovery analysis. Some of the WGA comparisons published in recent years [7,8,9,10,11,12,13,14,15] lacked in either the number of kits or cells analyzed per kit.

Short tandem repeats (STRs) exhibit much higher mutation rates than other genomic areas; are abundant in the genome; and serve as markers for genomic diversity in forensic, phylogenetic, and clinical applications. STRs make excellent candidates for evaluating the WGA artificial amplification noise thanks to their sensitivity to amplification errors in the form of slippage mutations that add or deduct whole repeat units. Similar in vivo behavior makes STRs a prolific source for heterozygous genomic sites that are used here to report on the balance between the two alleles in the outputs of each WGA kit.

Biezuner et al. [16] recently targeted ~1000 non-autosomal short tandem repeat (STR) loci using combinations of multiplex PCR with the Access Array microfluidics platform, covering seven kits and 125 cells but involving multiple rounds of PCR amplifications, cumulatively estimated at ~50 amplification cycles and covering only ~1000 loci on the X chromosome of male-derived cells. The duplex molecular inversion probe (DuMIP, also named the padlock probe) method for lineage discovery used here extends this comparison to a coverage of ~12,000 STR loci across the entire genome, reducing the amplification to ~20 PCR cycles and improving uniformity. This allows for a finer evaluation of the WGA kits preceding library preparation and the identification of the best WGA protocols for this new pipeline and similar sensitive downstream applications of WGA products (Figure 1).

## 2. Results

We started by generating a single H1 cell clone, as shown in Figure 1. Here, about two hundred single cells originated from one H1 cell were isolated into individual wells and amplified by eight WGA kits. We produced ~25 single-cell WGA samples with each kit. Duplex MIP pipelines were then applied to profile ~12,000 STRs from these single-cell WGA samples.

As the products of the single-cell WGA are ultimately utilized for DNA sequencing, imbalanced amplification results in the overexpression of some genomic regions and underexpression of others. With finite next-generation sequencing (NGS) coverage, underexpressed regions are dropped out of the results as other, more expressed regions take up most of the allocated reads. Targeted enrichment may mediate such effects to some extent but the dropout of alleles and loci would still be the result of imbalanced WGA, as evidenced by the improved performance of bulk samples.

Comparing the number of sequenced loci out of the 12,000 places, the top cells of Ampli1 and RepliG-SC had similar numbers of covered loci as the bulk samples (Figure 2). We also observed that the rates of failed cells with fewer than 500 loci were lowest for MALBAC and Replig-Mini, followed by PicoPlex and Ampli1.

The clonality and normal karyotype of the H1 cells allows the assumption that population-wide only two distinct alleles are present. This enables the confident identification of heterozygous loci in this comparison. Unlike heterozygous single-nucleotide variants (SNVs), heterozygous STRs are not always easily distinguishable; for example, two alleles of 29 and 30 AC repeats would result in highly overlapping stutter patterns following amplification [6]. To accurately report on allelic balance, the dataset was filtered for loci that satisfy two conditions, covered at over 30×, and genotyped with two easily distinguishable alleles, i.e., at least three repeat units apart when inspecting the global H1 cell population (Figure 3).

Ampli1 is notably the most balanced hit, followed by MALBAC and PicoPlex, all of which are PCR-based kits. While the H1-Bulk control depicts the most balanced profile, interestingly a positive bias is evident even here. This is due to the allelic identity assignment policy when reporting the proportion of the shorter STR genotypes, whereas longer STRs are selected against them in PCR [6].

In many cases where single cells are compared against each other via their WGA products, reproducible bias is preferred over a random bias. If only loci that are measured across the entire cell population were to be considered, random dropout would quickly diminish the available dataset. To evaluate this loci consistency quality of the WGA kits, we inspected the number of intersecting loci among cells in pairs, triplets, quartets, and quintets (Figure 4).

STR sensitivity to amplification error is well known and has been accurately calibrated under varying degrees of PCR amplification as part of the STR stutter genotyping tools used here [6]. The genotyping provides the simulated stutter pattern closest (correlations of no less than 0.95) to the measured histogram of STR lengths, with the simulation seed being the genotype itself and the number of simulated amplification cycles (values are equivalent to PCR cycles) serving to compare the amplification noise that results from the various WGA kits (Figure 5). Here, we note a clear separation between the MDA and PCR-based kits, whereby the latter are noisier, exhibiting the equivalent to 10~20 additional PCR cycles.

## 3. Discussion

Whole-genome amplification (WGA) is a major factor in single-cell genomic studies in terms of both price and faults. Choosing the appropriate kit to suit the experimental parameters is, therefore, crucial for the experiment’s success. Here, we compared eight commercially available WGA kits, conveying information on yield, genomic coverage, reproducibility, amplification noise, and allelic balance. We observed less amplification noise in the MDA = based WGA kits compared to the WGA kits that involve PCR. We measured extra amplification noise in Ampli1, Malbac, and PicoPlex equivalent to 10~20 additional PCR cycles. We think the main reasons behind the differences are: (A) the Phi29 DNA polymerase used in MDA kits has 10 times higher accuracy than PCR DNA polymerases used in PCR-based WGA kits; (B) MDA is an isothermal reaction, while PCR involves many rounds of denaturation over 90 °C. Allele balance is a crucial parameter for common single-cell WGA applications such as genotyping and detection of copy number variation. Using heterozygous STR loci and clonal H1 single cells, we measured the allelic balance on a large set of loci and cells. We observed Ampli1 as being the most balanced kit.

While it is clear from this and previous WGA comparisons that there is no single one-fits-all best WGA kit, we selected the Ampli1 as the most suitable kit for our STR-targeted retrospective lineage reconstruction platform due to its overall success rate, consistency of shared loci across cells, and allelic balance.

## 4. Methods

### 4.1. Clonal H1 Singlec-Cell WGA Samples

H1 human ES cells (WA01) and single-cell WGA material were banked in our lab by Tamir et al. [16]. Briefly, single H1 cells were picked and deposited in individual wells, cultured for ~2 weeks, then a single clone was selected to generate single-cell WGA materials according to the manufacturer’s manual for each WGA kit. A bulk positive control sample were also prepared from H1 cells with a DNeasy Blood and Tissue Kit from Qiagen(Hilden, Germany).

### 4.2. STR Target Sequencing

Several single-cell WGA samples from each WGA kit were selected for STR target sequencing as previously described [6,17]. The same 12,000 STR panel, named OM6, was used for this comparison. Briefly, 2 uL of WGA DNA from each sample was mixed with 8 uL OM6 hybridization buffer (final concentration of 0.8 fmol/µL OM6 MIPs, 1× Ampligase Buffer, and 0.9 M betaine) in 96-well plates. A hybridization reaction was run in a thermal cycler with a 100 °C lid temperature, then at 98 °C for 3 min, followed by a gradual decrease in temperature of 0.01 °C per second to 56 °C and incubation at 56 °C for 17 h. Then, 10 µL of gap filling mix (final concentration of: 0.3 mM each dNTP, 2 mM NAD freshly thawed from −80 °C, 1.1 M betaine, 1× Ampligase buffer, 0.5 U/µL Ampligase, and 0.8 U/µL Phusion^®^ High-Fidelity DNA Polymerase) was added to each well, incubated at 56 °C for 4 h, followed by incubation at 68 °C for 20 min and holding at 4 °C. Then, 2 uL exonuclease mix (final concentration of 3.5 U/µL Exonuclease I, 18 U/µL Exonuclease III, 4 U/µL T7 Exonuclease, 0.4 U/µL Exonuclease T, 3 U/µL RecJ_f_, and 0.2 U/µL Lambda Exonuclease) was used to remove all linear DNA via incubation at 37 °C for 60 min, 80 °C for 10 min, and 95 °C for 5 min. Then, 2 uL from each well was used as a template for Illumina library construction via PCR with a unique index. The individual libraries were cleaned using 1.0× Ampure XP beads. Concentrations were measured using a Qubit HS DNA kit. Here, 2 uL of each individual library was pooled and sequenced in MiSeq nano with 150 PE. Based on the number of reads for each sample obtained from MiSeq, volumes were calculated for equal molecular concentrations for each sample. Then, a balanced pool created by Echo550 was sequenced deeper with NexSeq550 for 150 PE.

### 4.3. Computational Analysis

The sequenced DNA samples were processed using the cell lineage discovery platform [5,17] to map and genotype the targeted STR loci. Briefly, STR-aware mapping of next-generation sequencing is performed against a custom reference genome that includes all the possible length permutations of the panel’s STR loci, resulting in repeat-number histograms per locus per sample. Those STR stutter histograms are then genotyped by comparison against simulated stutter distributions as described by Raz et al. [6], yielding the genotyped alleles, the proportion between them, and a measure of amplification-derived stutter noise in units equivalent to PCR cycles. GenomePlex and TruePrime ultimately failed (Figure 2) and the results reported here were from small datasets of the successful cases.

### 4.4. Data Access

Sequencing data generated in this study have been deposited to ArrayExpress (www.ebi.ac.uk/arrayexpress (accessed on 1 May 2022) under accession number E-MTAB-11711.

## Figures and Tables

**Figure 1 ijms-23-06161-f001:**
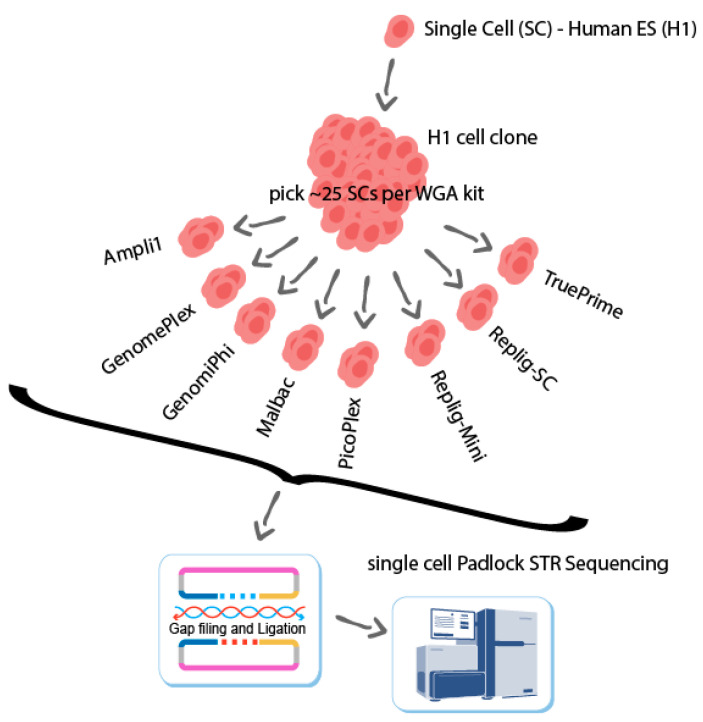
Experimental setup—a clone was generated from a single human ES cell (H1) that is considered normal and without known chromosomal aberrations. The clonal cells were dissociated and picked using the CellCelector (ALS) automated cell picker. The cells were then processed using different single-cell WGA kits (see [16] Methods section). Following single-cell WGA, the DNA samples were processed targeted for STR loci using the DuMIPs pipeline [6,17]. We analyzed the coverage (Figure 2), allelic balance (Figure 3), consistency (Figure 4), amplification noise (Figure 5), and mapping rates (Appendix A).

**Figure 2 ijms-23-06161-f002:**
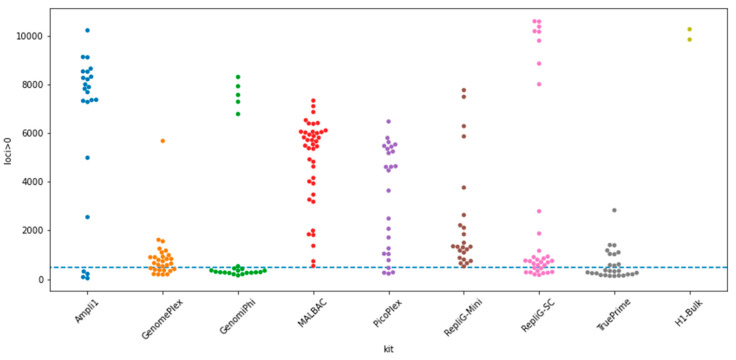
Loci sampled per cell (higher is better)—amplicon coverage per single cell per kit. Each dot represents a single-cell WGA sample, except for the bulk column, where each dot represents a cell bulk duplicate originated from the same cell line (H1). See also Appendix A Appendix A for additional normalizations.

**Figure 3 ijms-23-06161-f003:**
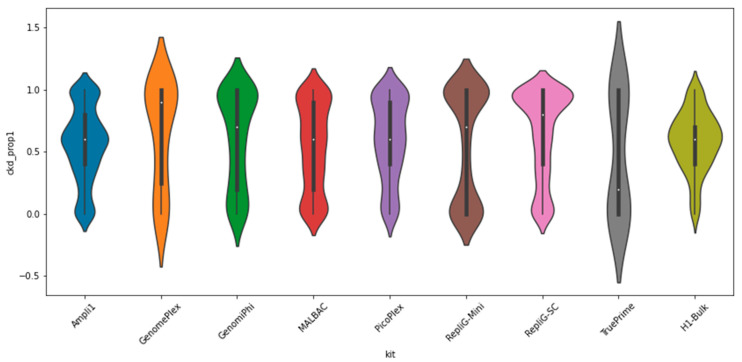
Allelic balance (closer to 0.5 is better)—distribution of proportions between alleles at heterozygous loci, where 0.5 indicates a perfectly balanced result, while 0/1 indicates the worst case. See also Supplementary Material Appendix A for an alternative view.

**Figure 4 ijms-23-06161-f004:**
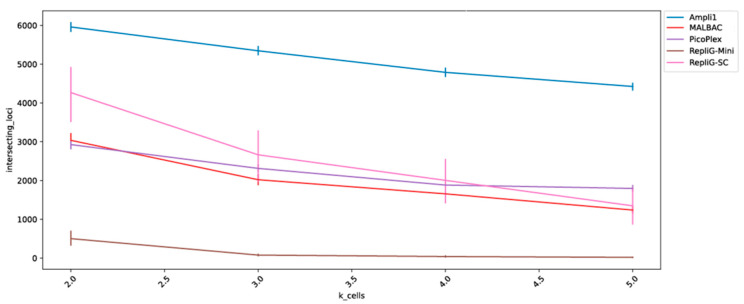
Loci consistently sampled in k cells (higher is better)—the number of intersecting loci is depicted on the Y axis for cell groups of 2–5 cells (X-axis). All samples were virtually sampled with 100,000 mapped reads and kits with less than 5 cells with at least that many reads were excluded.

**Figure 5 ijms-23-06161-f005:**
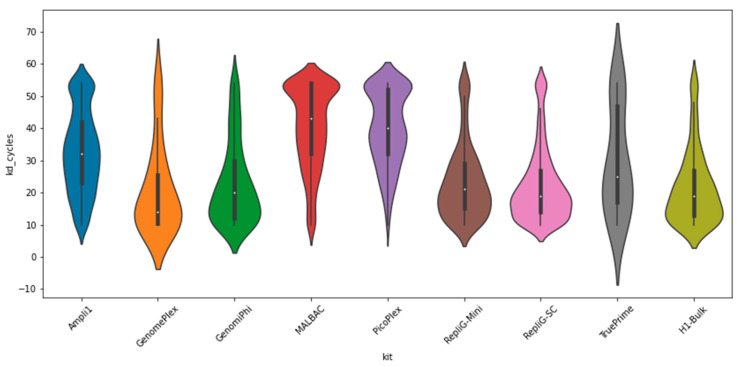
Amplification noise (lower is better). Amplification errors and quantification of the different single-cell WGA kits. Simulated STR stutter noise model was fitted for AC-type STR loci targets as part of the STR genotyping process [5]. The results clearly separate the MDA-based methods from the PCR-based one, which accumulates more relative stutter cycles, equivalent to up to 30 additional PCR cycles.

## Data Availability

Not applicable.

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
