# Peer review of "Whole-Genome Amplification—Surveying Yield, Reproducibility, and Heterozygous Balance, Reported by STR-Targeting MIPs"

_ijms, 2022, doi:10.3390/ijms23116161_

Round 1
Reviewer 1 Report
This manuscript reports a comparison between different commercially available WGA kits in order to verify which performs better and minimizes the introduction of biases that can act as confounding factors for the downstream applications. The paper is clear and easy to follow. The results are well exposed and may support research groups approaching to WGA.
Maybe it is a technical note more than a reasearch article.
Minor point:
- revise the style of the manuscript according to journal style (references in the abstract?)
- revise references format according to journal style
- avoid abbreviations' use in the abstract
- define the abbreviation at their first mention
- use the same abbreviation throughout the text, i.e. scWGA or SC WGA?
- check carefully the text for minor typo errors (resorted stay for reported, bias for biases, etc)
Author Response
- revise the style of the manuscript according to journal style (references in the abstract?)”; revise references format according to journal style.
A: Thanks, following the guidance from the editor, we revised the style and formation. - avoid abbreviations' use in the abstract; define the abbreviation at their first mention
A: Thanks, we fixed these abbreviations. - use the same abbreviation throughout the text, i.e. scWGA or SC WGA?
A: Thanks, we fixed this error.
- check carefully the text for minor typo errors (resorted stay for reported, bias for biases, etc)"
A: Thanks, we fixed these errors.
Reviewer 2 Report
This is a clearly described study that will help scientists for future experimental designs.
Minor suggestions:
- Giving references in the abstract is rather unusual.
- Please explain all abbreviations used in the text when you introduce them first, e.g. SC, MIP.
- Lines 28 to 35: please give more references
- Please check your reference list. Names of authors are missing.
- Figure 4: You should remove all excluded kits from the figure legend. Keeping them in the legend, but not showing them on the graph is confusing.
Best regards.
Author Response
- Giving references in the abstract is rather unusual.
A: Thanks, we removed the references from abstract - Please explain all abbreviations used in the text when you introduce them first, e.g. SC, MIP.
A: Thanks, we fixed the abbreviations. - Lines 28 to 35: please give more references
A:Thanks, we added relevant references. - Please check your reference list. Names of authors are missing.
A: Thanks, we fixed these errors. - Figure 4: You should remove all excluded kits from the figure legend. Keeping them in the legend, but not showing them on the graph is confusing.
A: Thanks, we updated Fig4 as you suggested.